



**Source apportionment of atmospheric mercury in the remote marine atmosphere: Mace**
**Head GAW station, Irish west coast**

Danilo Custodio[1], Ralf Ebinghaus[1], T. Gerard Spain[2] and Johannes Bieser[1]
[1] Helmholtz-Zentrum Geesthacht, Institute of Coastal Research, Max-Planck-Str. 1, D-21502 Geesthacht, Germany.
[2] National University of Ireland, Galway, Ireland.
Abstract
We examined recent atmospheric mercury concentrations measured with a high temporal resolution
of 15 min. at Mace Head, a GAW station on the west coast of Ireland. We attributed a direct
contribution of 34% (0.44 ng m$^{-3}$) to primary sources. Additionally, a steep decline (0.05 ng year$^{-1}$) in
mercury concentrations was observed between 2013 and 2018.
Using a stereo algorithm we reconstructed 99.9% of the atmospheric mercury. A conservative
analysis demonstrated no decreasing of TGM associated with atmospheric species typically used as
tracers for oceanic emissions. The results show that the atmospheric mercury mass is mainly loaded in
a baseline factor with an on-going decline. Moreover, we exploit temporal variation and wind pattern
effects in the measured atmospheric species, the results show that the diurnal variation and
seasonality in TGM observed in Mace Head is closely related to other species linked to primary sources
and can be explained by transport from continental areas.


**1. Introduction**
Atmospheric mercury is a bioaccumulative, toxic pollutant with the potential to be transported over
large distances that poses a significant public health and environmental problem (WHO, 2007).
Despite efforts by governments and international agencies as well as the private sector to reduce
mercury release into the environment, current environmental levels are often still of concern.
Atmospheric mercury is emitted from both natural and anthropogenic sources as well as through
recycling of past emissions. Natural sources are comprised of release from volcanoes, weathering of
rocks, forest fires and oceanic emissions. Anthropogenic sources are related to fossil fuel combustion,
cement production, industrial activities, mining and municipal or medical waste incineration. Mercury
is also reintroduced into the atmosphere through natural processes such as oceanic evaporation after
reduction of inorganic oxidized Hg in anaerobic environments, which leads to global cycling of this
element (Corbitt et al., 2011; Streets et al., 2011). The source contribution, as well as the life-time of
atmospheric mercury, is only roughly estimated.
The 2018 Global Mercury Assessment (UN, 2018) reveals that primary anthropogenic mercury
emissions into the air are 2220 t/y, also indicating an increase of 20% from such sources in recent





years. The 2018 UNEP Report (AMAP/UNEP, 2018) presents an inventory for the year 2015, which
indicates that the greatest atmospheric mercury emissions resulted from combustion of fossils fuels,
mainly coal. While mercury in the atmosphere is chemically inert, once released into this environment,
all sources are of concern.
To compile a global assessment based on inventories requires a number of assumptions and
generalizations (AMAP/UNEP, 2018). Several discrepancies are observed in the mass balance-based
estimation: there can be large differences between estimates, and it is important to recognize that
there are sources of error in all methods for estimating mercury emissions.
Here we report concentrations of atmospheric mercury (TGM: total gaseous mercury) measured from
January 2013 to March 2018 at Mace Head. Mace Head station is located within the central North
Eastern Atlantic region and based on a GEOS-Chem simulation it is one of the most influenced
region by a decreasing mercury trend in ocean surface water, according to Soerensen et al.

51    (2012).

Using the relationship between mercury and other chemical atmospheric trace species ($O_3$, CFC-12,
$CCl_4$, $N_2O$, $CH_4$, $CHCl_3$, CO and $H_2$) and meteorological data (wind speed and direction), we performed
a mass balance to reconstruct atmospheric mercury. Solved by positive matrix factorization, the total
mercury mass was distributed into four different factors, classified as baseline, combustion, oceanic
and a fourth factor and then each of them was assessed for source trends.
Time series analysis of atmospheric mercury concentrations at Mace Head were already
reported by Weigelt et al. (2015) and Ebinghaus et al. (2011).
In this work we apply a new approach for source apportionment and extend the time series
analysis up to March 2018.
**2. Experimental Setup**

**2.1. Sampling site and analytical methods**
Mace Head atmospheric research station is located on the west coast of Ireland at $53.33^0$N and $9.54^0$W,
55 km from Galway (80,000 inhabitants), the nearest city with significant industrial activity. It is a GAW
baseline station, exposed to the North Atlantic Ocean and is an ideal location to study both natural
and anthropogenic trace constituents in marine and continental air masses (Stanley et al., 2018).
In addition to atmospheric mercury, meteorological parameters are routinely monitored
(https://www.met.ie/). Atmospheric CFC-11, CFC-12, $CHCl_3$, $CCl_4$, $N_2O$, $CH_4$, CO and $H_2$ are measured
(Figure S1) as part of the AGAGE project (https://agage.mit.edu/).





TGM is monitored by an automated dual channel, single amalgamation, cold vapour atomic
fluorescence analyser (Tekran Analyzer Model 2537B, Tekran Inc., Toronto, Canada) described by
Ebinghaus et al. (2011).
The air-sampling inlet is located on a tower at 10m agl (18m amsl). Air is sampled at a flowrate of 1
L/min through unheated PTFE tubing (1/4" O.D.) to the instrument, which is located in an air-
conditioned laboratory. As reported by Weigelt (2015), a PTFE pre-filter (pore size 0.2 mm) at the inlet
of the instrument protects the sampling cartridges from contamination by particles. The device is
operated with a temporal resolution of 15 minutes, calibrated every 25 hours using an internal mercury
permeation source. The device has a detection limit of ~0.1 ng m$^{-3}$ (Weigelt et al., 2015).
Furthermore, wind streamlines for near surface level conditions were assessed from
https://earth.nullschool.net/ and long-range transport of air pollutants was calculated using the
HYSPLIT model (Draxler and Rolph, 2003) from NOAA (National Oceanic and Atmospheric
Administration).

**2.2. Source assessment / Probability mass function**
Apportionment of atmospheric species is often performed by receptor models that are based on the
mass conservation principle:

89          The inclusion of the potential rotated infinity matrices transformation produces factors that

appear to be closer to realistic chemical profiles of sources:
$$x_{ij} = \sum_{k=1}^{p} g_{ik} f_{jk}$$
$i=1,2,…,m \quad j=1, 2…. n$ (1)

where $x_{ij}$ is the concentration of the species $j$ in the $i$th sample, $g_{ik}$ is the contribution of the factor
(associated to a source) $k^{in}$ in the $i$th sample and $f_{jk}$ is the concentration of the species $j$ in factor $k$ as
*presented* by Paatero and Hopke (2003) and described by *Comero et al. (2009)*. This equation can be
solved by the probability mass function in *positive matrix factorization* (PMF) (Paatero and Tapper,
1994) with the Multilinear Engine (ME-2) developed by Paatero (1999) and implemented in Version 5
of the US EPA PMF (https://www.epa.gov/air-research/positive-matrix-factorization-model-
environmental-data-analyses).
In this study, PMF was applied to the Mace Head dataset with an hourly time resolution for the period
2013 to 2018. The results were constrained to provide positive factor contribution. The uncertainty
input in the matrix was estimated based on the analytical accuracy of each individual species.
PMF is a stereo algorithm where analytical data sets are combined to create fingerprints and the profile
is used to assess the contribution of each source based on the mass load, also providing a robust
uncertainty estimation and source diagnostics. The method provides a better solutions and time
resolution of sources than principal component analysis (PCA) (Huang et al., 1999) or chemical mass





balance (CMB) since PMF can generate source profiles ("learning algorithm") and let input of
uncertainties which allow individual treatment of matrix elements.
In the PMF the weighted factorization regression analysis is based on positive rotable factorization of
non-singular matrix T;
$X = F G + E = G T T^{-1} F + E = \overline{G}\,\overline{F} + E,$ (2)
where the new rotated factors are
$\overline{G}$ = G T and $\overline{F}$ = T$^{-1}$ F as reported by Comore et al. (2009), then the factors are no-negatively
constrained.
Factors contributions are chosen on the basis of a matching strength score by using a form of discrete
correlation. At the first interaction any matches which have the highest matching strength for
primitives mass reconstruction that formed them are immediately chosen as reconstructed. Then, in
accordance with the uniqueness constraint, all other matches associated with the primitives that have
been formed for each chosen match are eliminated from further consideration. This allows further
matches that were not either previously accepted or eliminated to propagate the process of PMF to a
satisfactory solution if the propagation converges.

**3. Results**
Time series of TGM concentrations composed of 48,914 hours of measurements covering the period
from January 2013 to March 2018 are given in Figure 1. Concentrations range from 0.9 to 3.3 ng m$^{-3}$,
displaying a central tendency of 1.3 ± 0.2 ng m$^{-3}$. TGM concentrations in the northern hemisphere have
been decreasing in recent decades (Ebinghaus et al., 2011; Slemr et al., 2003). For instance, Ebinghaus
et al. (2011) reported a decline  trend of 0.028 ± 0.01 ng m$^{-3}$ yr$^{-1}$  between 1996 to 2009. Account the
more recent years (1996 to 2018), this decline continued with approximately 0.025 ± 0.04 ng m$^{-3}$ yr$^{-1}$,
figure 2. This observation could reflect a trend in global emissions, as mercury, roughly, has an
atmospheric lifetime of 0.5 to 1 year (Holmes et al., 2006; Lindberg et al., 2007; Si and Ariya 2018). The
increasing improvement of manufacturing processes involving mercury and regulations limiting the
emissions from coal-fired power plants since the 1980s (Hylander  and Meili, 2003; Pirrone et al., 2009)
could be a possible reason for this observed decline at Mace Head. Jiskra et al. (2018) report the Hg$^{0}$
uptake by vegetation as an alternative mechanism for driving mercury depletion in the Northern
Hemisphere atmosphere over the past 20 years.
However, this decreasing trend is inconsistent with the increased emissions from 1990 to 2015, as
indicated by anthropogenic Hg emission inventories (e.g., UN, 2018 and AMAP/UNEP, 2018).
**3.1. Temporal and wind pattern effects in mercury concentrations**


Plots of TGM as a function of wind speed and direction can be seen in Figure 3 as well as the polar
frequency plot of wind direction. Concentrations of mercury are higher when winds come from the
east (continental air masses) and lower for winds from the west and northwest (Atlantic air masses).
The higher concentrations to the east are likely to be influenced by urban agglomerations, such as in
Galway, Dublin or even the UK and continental Europe. These higher levels observed to the east are
associated with relatively strong wind speeds of 15ms$^{-1}$, which could indicate a relatively distant
source. Furthermore, an increase of TGM with strong winds of 20 ms$^{-1}$ was observed, indicating sources
at further distances in air masses coming from westerly and south-westerly directions. 96-hour back
trajectories show that these high TGM concentrations at Mace Head were affected by air mass
transport from the Iberian Peninsula and long-range transport from North America.
Higher mercury concentrations under the influence of easterly and strong westerly/south-
westerly winds closely resemble those of other pollutants that are also closely linked to
anthropogenic emissions, such as carbon monoxide, and suggest TGM enrichment from
continental air masses.
The polar plot shows low concentrations of mercury associated with strong and weak winds
coming from the North Sea and nearby land air masses, with  in < 10 m s$^{-1}$.
The diurnal cycle of elemental mercury (Hg$^0$) has been discussed extensively (Laurier et al., 2003;
Weiss-Penzias et al., 2003; Laurier and Mason, 2007; Xia et al., 2010; Obrist et al., 2011; Moore et al.,
2013; Wang et al., 2014; Ci et al., 2015; Wang et al., 2017; Castagna et al., 2018, Jiskra et al., 2018).
Kalinchuk et al. (2019) reported solar radiation-driven increase and decrease of mercury
concentrations in the Sea of Japan and in the Sea of Okhotsk, respectively. They assumed that the
decrease in Hg$^0$ concentrations in the marine boundary layer during daytime is mainly caused by its
oxidation, catalyzed by active halogen species (mainly by atomic bromine radicals), which are released
from sea salt aerosols as Br$_2$ and could be transformed into reactive forms as a result of photolysis
(Holmes et al., 2009; Sprovieri et al., 2010; Mao and Talbot, 2012; Moore et al., 2013; Si and Ariya,
2018). However, the absence of a diurnal cycle for mercury is reported in several studies and more
research should be done to confirm the catalytic photolysis oxidation, as large uncertainties exist in
the gas-phase reaction of mercury (Si and Ariya, 2018).
With a standard electrode potential (E$^0$) of +0.85 V and a kinetic coefficient of reactivity of <9.8 × 10$^{-13}$
to 2.1 × 10$^{-12}$ cm$^3$ molec$^{-1}$ s$^{-1}$, at 1 atm and 298 K (Khalizov et al., 2003; Shepler et al., 2007; Subir at
al., 2011; Sun et al., 2016), Hg$^0$ is quite a stable vapour gas, and a significant daily mass depletion by
photooxidation is very unlikely.
Seasonality and diurnal patterns for mercury concentrations at Mace Head have been detected, but
similar patterns were observed for CO. As presented in Figure 4, wind direction was a driving factor for
diurnal cycling of TGM at Mace Head as well as for CO and CHCl$_3$. Winds from the east (land breezes)



showed sharp increases of TGM, CO, CFC-12 and CCl$_4$ (figure 3 and Figure S3). Conversely, an increase
of CHCl$_3$ in offshore winds (sea breezes) was observed.
Mace Head is mostly influenced by air masses from the Atlantic Ocean, however, as a coastal site can
be affected by on-shore breezes blowing from land to the North Atlantic. Daily fluctuations of wind
speed and direction in coastal areas are a result of differences in air pressure created by the different
heat capacities of water and dry land (Yan Y.Y., 2005).
Decrease of atmospheric mercury concentrations during warm periods has often been linked to
increased Hg$^{2+}$ by catalytic mercury oxidation in the surface layer of the sea due to several chemical
and biological processes, mainly controlled by solar radiation (Kalinchuk et al., 2019 and references
therein). Si and Ariya (2018) and references therein reported maximum oxidation of mercury in
summer based on several atmospheric models but failed to reconstruct observed summer depletion
of atmospheric mercury at monitoring sites in North America and Europe. Furthermore, deposition
models could not predict the observed large seasonal variability of either Hg oxidation or wet
deposition flux (Travnikov et al., 2017).
Figure 4 shows that the decrease of TGM during summer is closely related to CO depletion in this
season.
In addition, it was observed similarity among TGM depletion during summer, enhancement during
autumn and seasonality of chloroform (CHCl$_3$). Decreased emissions of CHCl$_3$ from seawater or more
intense depletion by photooxidation during summer may be possible explanations. It should be noted
that any photochemical pattern of those species must be considered with caution because CHCl$_3$ is a
short-lived species (lifetime ~0.5yr), mainly produced in the ocean by biological processes that follow
a different oxidation pathway than mercury (Khalil and Rasmussen, 1999). It should also be noted that
wind pattern differences were observed within one year for Mace Head: strong winds during winter
predominately comes from the sea, and relatively calm winds during summer (Figure S2). This should
also be reflected in the observed seasonality of TGM concentrations.
The results obtained during this study show that the seasonality in TGM observed in Mace Head is
closely related to other species linked to primary sources and can be explained by transport from
continental areas.

**3.2. Source apportionment**
Figure 1 shows the set of four factors reconstructing atmospheric mercury concentrations obtained
from the PMF solution. As reported by Henry (1991), the first set of natural physical constraints of the
system to be considered in any approach for identifying and quantifying source mass contributions
must be the reconstruction of the original data set by the algorithm—that is, the solution must explain
the observations. Figure 5 shows that the sum of the predicted elemental mass contributions for all


sources is almost the same as the total TGM measured. Lower reconstruction performance was
observed in particular for concentrations higher than 2 ng m$^{-3}$, which make up 0.44% of the
observations.
The first factor with a loading of 66% of TGM mass (0.88 ng m$^{-3}$) was labelled as baseline because it
does not show any wind pattern, carries high loads of long-lived species such as CFCs and low loads of
CO or sea-borne trace gas species. The PMF results show a statistically significant decrease in the
baseline factor that could explain almost all of the trend changes in atmospheric mercury. This suggests
a major decrease of anthropogenic inputs on a global scale. Slemr et al. (2011) reported a worldwide
trend of atmospheric mercury, showing an equally strong decrease in the northern and southern
hemispheres, which supports the argument of baseline-driven TGM decline.
According to Streets et al. (2011), anthropogenic Hg emissions in the USA and Europe decreased by
20% and 40%, respectively, from 1990 to 2008. However, emissions on a global scale, particularly from
East Asia, are poorly reported (UN, 2018), even for most of the countries that are signatories of
Minamata convention (UN, 2019). Moreover, the total emissions from small scale artisanal gold mining
are highly uncertain estimates.
Another possible explanation for the declining trend may be the Hg$^0$ atmospheric life-cycling reduction
due to atmospheric acidification caused by $CO_2$ potential (E°) to force elemental mercury oxidation.
As reported by Slemr et al. (2011) and references therein, increase in the atmospheric reactivity can
induce large decreasing trends in the concentration of many long-lived substances. Clerbaux and
Cunnold, (2007) did not observe lifetime changes for halogenated and other greenhouse gases,
however, changes in oxidation rates of elemental mercury in the atmosphere could follow different
kinetics. Furthermore, the increasing UV radiation and the shifting solar radiation to shorter
wavelengths could also intensify the oxidation of elemental mercury into Hg$^{2+}$ (IPCC, 2007; Qureshi et
al., 2010). Jiskra et al. (2018), on the other hand, hypothezise that increased vegetation uptake could
be a reason for decreasing atmospheric mercury concentrations in recent years.
A second factor that contributes to mercury with 0.27 ± 0.13 ng m$^{-3}$ (21 %) and is characterised by a
high load of CO and labelled as combustion. A decreasing trend was observed in this factor, but this is
a more complex case because a higher load of Hg in the combustion factor could be strongly influenced
by wind direction, as shown in Figure 6. For the potential seasonality, significant trends are also difficult
to establish due to the relatively short time series. The Global Mercury Assessment inventory (UN,
2018) estimates the contribution of combustion sources to atmospheric mercury at 24%.
The wind patterns for the baseline, combustion and sea factors (discussed below) as displayed in the
polar plot of Figure 6 indicate an interpretation of the PMF profile with "combustion" being mostly
associated with easterly transport, "sea" being linked to north-westerly and south-westerly winds. The
"baseline" factor does not correlate with any significant wind patterns.





The seasonality observed in the factors fingerprinted by $CHCl_3$ and CO (Figure 7) should, however, be
considered with caution because those short-lived species ($CHCl_3$ 4-5 months and CO 1-3 months) have
lifetimes that vary by season, which can dampen mercury load into its factor during summer. However,
no seasonality was observed for the baseline factor, linking lower concentrations of mercury in the
warm season mainly to transport or evasion patterns and less to deposition by oxidation.
Human activity has substantially increased the ocean mercury reservoirs and consequently the fluxes
between the ocean and atmosphere (Strode et al., 2007; Smith-Downey et al., 2010).
The residence time of mercury in the ocean is substantially longer than in the atmosphere, ranging
from years to decades or millennia (Strode et al., 2007; Primeau and Holzer, 2006). Acidification of
oceans, climate change, excess nutrient inputs, and pollution are fundamentally changing the ocean's
biogeochemistry (Doney, 2010) and will certainly also influence mercury ocean-air fluxes (Slemr et al.,
2011). The extent, however, and even the direction of the change is unknown.
Mason et al. (2012) estimate that global oceanic $Hg^0$ evasion to be comparable to anthropogenic
emissions, and Sunderland and Mason (2007) attributed the mercury emitted from seawater in the
North Atlantic to the legacy of 20[th]-century anthropogenic sources in Europe and North America.

This study shows an oceanic contribution of 13% ($0.17 \pm 0.07$ ng m$^{-3}$) to atmospheric TGM at Mace
Head station. Based on atmospheric mercury concentration trends in the subsurface seawater
Soerensen et al. (2012) predicted a decrease of approximately 0.045 ng m$^{-3}$ yr$^{-1}$ of oceanic mercury
emissions into the air over the North Atlantic. They also argued, based on cruise data, that the decrease
of oceanic emissions is forcing the atmospheric trend. In this study, based on the PMF solution, we
found no evidence for a decreasing mercury load in the oceanic factor, which could be traced by $CHCl_3$
and $CH_4$ concentrations.
A fourth factor with a high load of $O_3$ and CO was found by the PMF solution which appeared to be
irrelevant for the mercury mass balance, as its load was just 0.003 ng m$^{-3}$. However, for atmospheric
mercury concentrations higher than 2 ng m$^{-3}$ this factor had a load of 0.57 ng m$^{-3}$. In addition, with 0.53
ng m$^{-3}$ the mercury load in the combustion factor for concentrations higher than 2 ng m$^{-3}$ is twice as
high as for concentrations below 2ng/m$^{-3}$ in this sector. (Figure 8).
Moreover, we find from the PMF solution that the decrease of atmospheric mercury is linked less to
oceanic emissions and is explained mainly by a baseline factor with a low load of short-lived species
with significant anthropogenic sources, such as CO and $O_3$, as well as a low load of sea trace species,
such as $CHCl_3$ and $CH_4$.
On the other hand, a decrease in mercury is observed in the factor with high loading of long-lived
species such as CFCs. However, the presented solution for apportionment of atmospheric mercury has
restrictions and requires further consideration, as the mercury sources are complex and numerous,
and merely a few source tracers were used in this study.



**4. Conclusions**

This study presents a comprehensive source assessment of atmospheric mercury measured
at Mace Head, a baseline station with a long-term decreasing trend of TGM. Positive matrix
factorization was applied to a set of atmospheric mercury data from 2013 to 2018 with high
temporal resolution. The profiles of source factor contributions indicate that baseline (0.86 ng
$m^{-3}$, 66%) and combustion processes (0.27 ng $m^{-3}$, 21%) are the controlling factors of mercury
in the atmosphere at this remote coastal measurement location. The high load of mercury in
the baseline factor reflects the relatively long lifetime of this species in the atmosphere.
Biogenic activities in the ocean were identified as another primary source, contributing 13 %
(0.17 ng $m^{-3}$).
Therefore, based on the analysis of temporal changes in the sources, no decreasing in the
oceanic factor in the period of this study could be detected. The decrease in atmospheric
mercury concentrations was linked to the baseline factor. Source contributions by wind sector
were also exploited, based on directional wind dependence of source loadings from the PMF
analysis. The patterns are also consistent with the location of the sources: oceanic sources
coming from the west (Atlantic) and anthropogenic sources coming from east (Europe) of
Mace Head. Furthermore, more extensive and detailed descriptions concerning mercury
sources is needed to confirm and evaluate the reported trends, which then can have great
relevance for policy and regulations in light of the Minamata convention.

**Acknowledgments**
Acknowledgements. This work was funded by the iGOSP ERA-PLANET and E-SHAPE
*"EUROGEOSS" Showcase* projects. The author acknowledges the Mace Head
Observatory for all data provision. The authors gratefully acknowledge the NOAA
Air Resources Laboratory (ARL) for the provision of the HYSPLIT transport and
dispersion model and READY website (http://www.ready.noaa.gov).

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

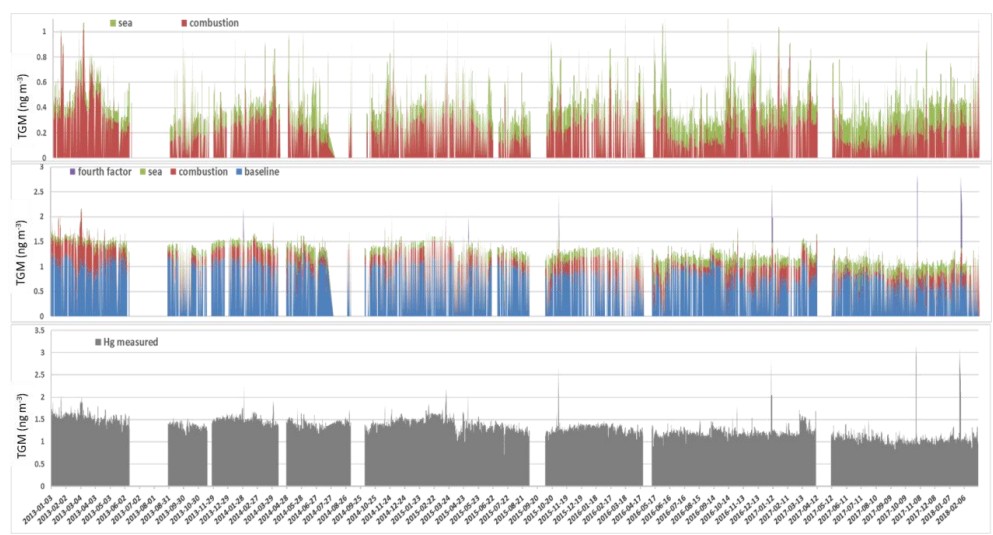

Figure 1. TGM hourly variations measured at Mace Head, from 2013 to 2018 (bottom), time series of mercury
attributed to each factor (center) and time series of sea and combustion only (top).

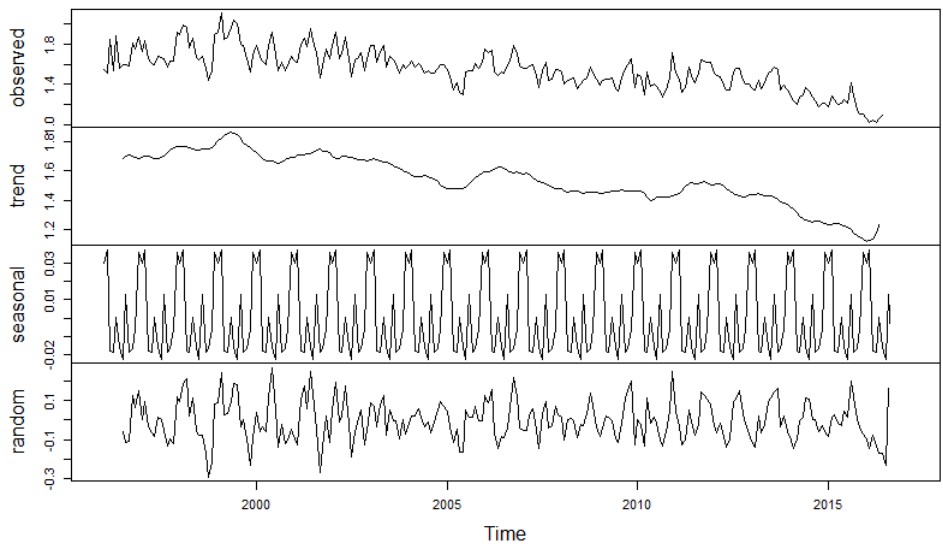

Figure 2. Time series decomposition of TGM (monthly averages) measured at Mace Head from 1996 to February
2018. * TGM in ng m$^{-3}$.

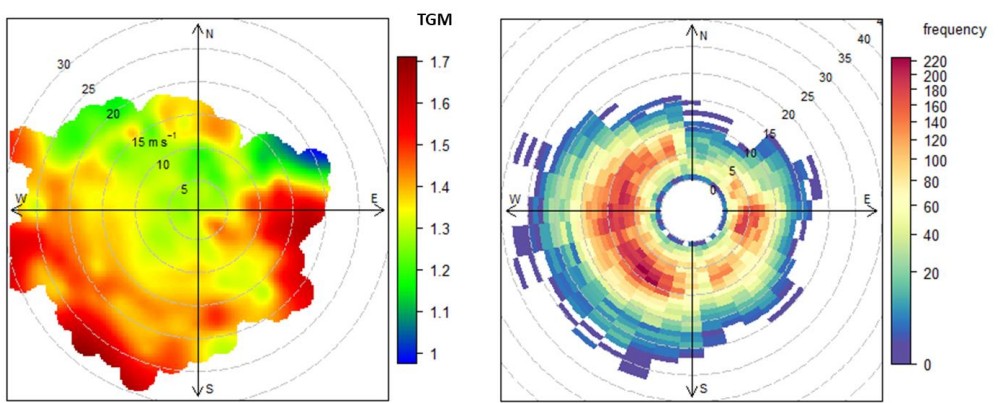

Figure 3. Polar plots for TGM (left) and polar wind frequency (right) at Mace Head. * TGM in ng m$^{-3}$ and wind
speed in ms$^{-1}$.

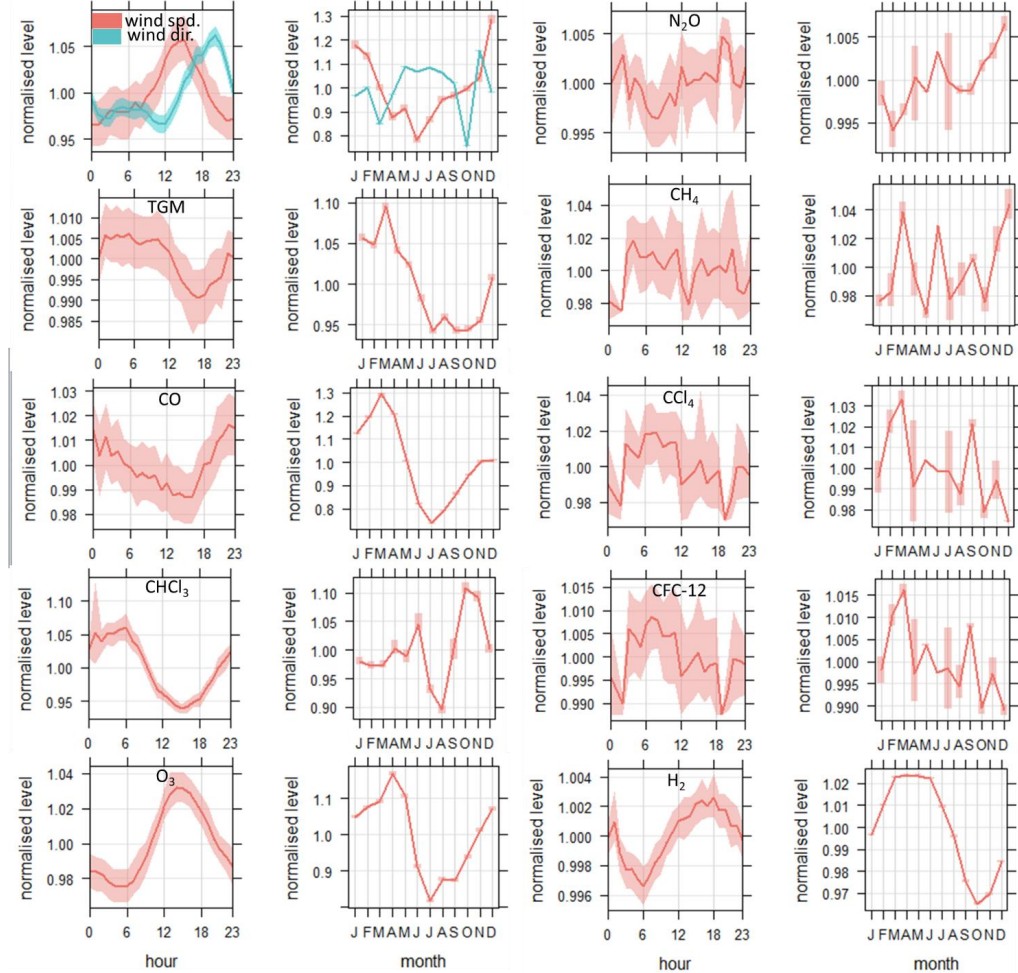

Figure 4: Diurnal cycle and seasonal cycle of mercury and species loaded in the PMF matrix. The shaded areas
are the 95% confidence intervals in the mean. *Wind direction is normalised with 0° defined as North.



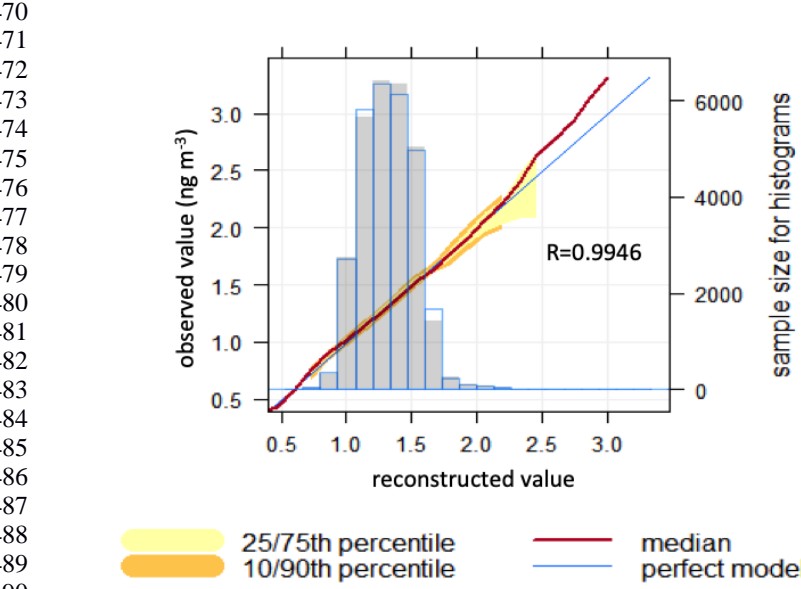

Figure 5: Correlation among total elemental mercury measured and mercury reconstructed by the PMF solution.

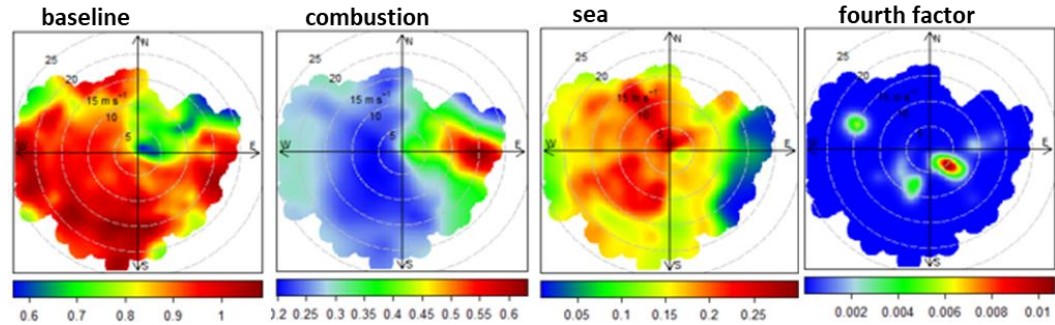

Figure 6. Polar plots for the factors obtained in the PMF solution. The plots show variations of mercury (ng m$^{-3}$) loaded in each factor as a function of wind direction (°) and speed (ms$^{-1}$).

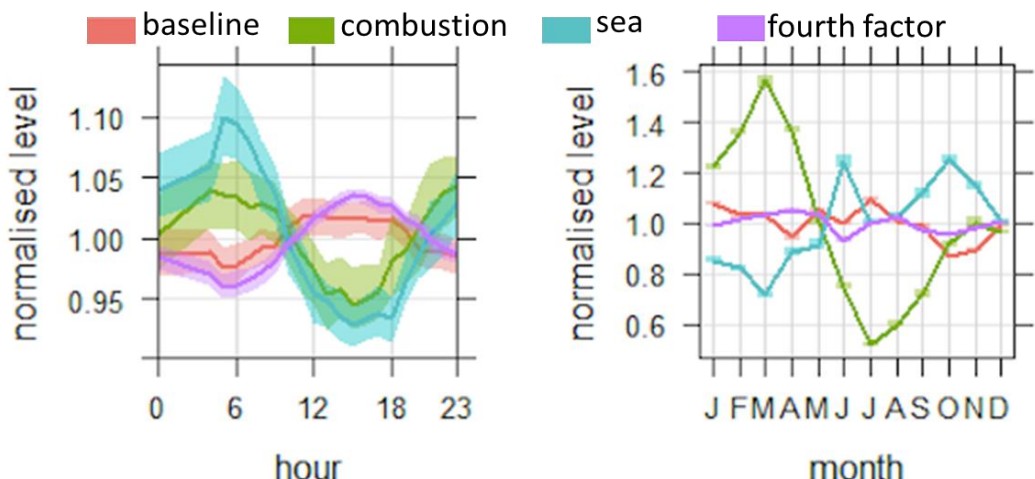

Figure 7: Mean and 95% confidence interval in mean of diurnal and seasonal cycle of four PMF factors.

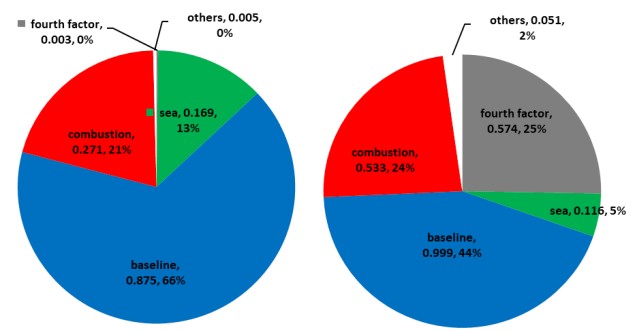

Figure 8. Average contribution (ng m$^{-3}$ and %) of Hg$^0$ factors for Mace Head from 2013 to 2018 (left) and mass
closure for mercury concentration greater than 2 ng m$^{-3}$ (right).