# Peer review of "Source apportionment of atmospheric mercury in the remote marine atmosphere: Mace Head GAW station, Irish west coast"

_Atmospheric Chemistry and Physics, 2020_

## Referee Comment (RC1) · Anonymous Referee #1 · 30 Mar 2020

Mace Head, GAW station provides valuable long-term observational data for atmospheric mercury in a coastal region in middle-latitudes. The data has been extensively interpolated for source appointment and atmospheric trend attribution. In this study, the authors utilized a receptor modeling technique for source apportionment that involves other chemical atmospheric trace species and meteorological data. This is a new addition to mercury trend analysis and the conclusions are generally reasonable. Clearly it merits publishing in ACP, but not in the current form. My major concern is the organization of the article. The authors made conclusions and speculations all through the results and discussion section. But some of them are not fully supported and seem hasty. I suggest reorganizing the paper to separate the result and discussion

sections. By this mean the author can first present all the results, and then interpret them, especially their interactions as they are so closely associated (e.g. the results of hourly, monthly, and annual cycles and their associations with other chemical tracers and meteorological data). My detailed comments are as follow:

Line 136-137: this conundrum has an explanation in Zhang et al. PNAS, https://doi.org/10.1073/pnas.1516312113.

Line 167-169: It's not clear how the standard electrode potential or the kinetic coefficient of reactivity is translated to the conclusion that "Hg0 is quite a stable vapor gas, and a significant daily mass depletion by photooxidation is very unlikely".

Figure 4: wind direction has no y-axis.

Line 194: A specie with a lifetime of $\sim$0.5 yr is not a short-lived one.

Section 3.1. The authors made some conclusions in this section, e.g. line 169-170 and 199-201. These conclusions seem unreliable and hasty. Why not waiting after presenting the PMF results?

Figure 5: The histogram of reconstructed value is not helpful. I suggest showing that of error.

Line 224-233: "atmospheric acidification"? Is it actually "atmospheric oxidation"? Also, this paragraph reads very confusingly with so many turns around.

Line 215-233: I suggest cutting the length of such speculations, they are very long and basically a review of past results. What new information is revealed by the author's own data and analysis?

Line 260: It's risky to call this component as oceanic contribution as i) the fraction is very low; and 2) the baseline fraction may contain a contribution from the global ocean evasion fluxes. I would suggest using the term "nearby ocean contribution".

Many orphan sentences throughout the article. I suggest combine them with nearby

paragraphs.

---

## Referee Comment (RC2) · Anonymous Referee #2 · 30 Mar 2020

Overall this is a good manuscript. The results and analysis are valuable additions to the scientific community. I have provided some technical suggestions to hopefully improve the manuscript and some editorial suggestions to improve the readers experience. My recommendation is that this manuscript be accepted with revisions.

Technical suggestions: 1. Page 2 line 48. You use data from January 2013 to March 2018. I would suggest that when reporting annual trend data that you do not include 2018 in that data given that you do not have a full year and it may skew the results. Of course monthly reporting works but be mindful with annual reporting. 2. Page 3 line 72 – please add in a description of how the data was quality assured. What is the

level of completeness of the data used in the analysis. This is very important when the data is compared to other data sets. 3. Page 3 line 75 - please mention whether there was a filter on the outside inlet and if there is a rain shield etc. This is important again for comparison with other data sets. You can refer to other papers as appropriate if described there. 4. Page 3 lines 89-97 – I don't understand this very much. Can you give a small sentence with what this analysis will provide to the data in layman's terms? 5. Page 3 line 101 – it would be interesting to see the information of the accuracy of each of the analytical species to see where the largest error occurs. If doable, a table in supplemental information? 6. Page 3 line 104 – "the method provides a better solutions" – what do you mean by that? Solution to what? Be specific. 7. Page 4 lines 106-120 – The reader would benefit from a short explanation in simple terms on what one gets from this type of analysis 8. Page 4 line 125 – what do you mean "central tendancy"? 9. Page 4 line 128 – does this trend include the Jan – march 2018 data and should that be included? 10. Page 4 line 127-128 – any thoughts on why the decline is so consistent? Have you looked to see if there is a change over time (the data from 1995 is available) to see if the rate of decline changes year over year or periodically? Maybe look at the trends in 5 year trends similar to how emissions are reported (i.e. each 5 years)? 11. Page 5 line 164 – in the reported studies of diurnal cycles, were they coastal sites like Mace Head? 12. Page 5 lines 167-169 –Photoxidation may be unlikely but you have to consider the atmospheric conditions that lead to enhanced photooxidation at various locations and not only consider the properties of Hg0 13. Page 7 line 214 - "the PMF results show a statistically significant decrease in the baseline factor" – I am not sure I see that in what is presented. Can you make it clearer please? 14. Page 7 line 220 – you mention the decreases but why not include the increases during that time as well. 15. Page 7 line 221 – I don't think the point about countries being valid signatories is relevant. Initial reporting were only due in December 2019 and full reporting is only due 2021. So, this statement isn't fair under the context of the Minamata Convention. 16. Page 7 lines 224-232 – Maybe have a look at some trend data from speciated Hg monitoring and use that to comment

on this paragraph. 17. Page 7 line 238 – But you have far more data available from this location...why not use that? 18. Page 8 line 244 – CHCl3 and CO are not mentioned in Figure 7 19. Page 8 lines 244-255 – the explanations are a little all over the place. Perhaps a more organized discussion could be done here so the reader can follow more easily. 20. Page 8 line 264 – "...is forcing the atmospheric trends". Do you mean rather than emissions? If so, please state that so its clear 21. Page 8 line 268 – maybe include the % here to stay consistent with the other 3 factors explained. Line 269, 0.57ngm-3 is very big! Can you put that in a percent and also offer more insight as to why. Editorial suggestions: 1. Page 2 line 56th. It's a little odd that you don't give this factor an identification as you have done for the others. I would suggest naming it something relevant to what its looking at rather than saying 4th factor which doesn't define it at all. 2. Page 3 line 81 – remove "furthermore" this adverb is not appropriate here. 3. Page 3 lines 102-104 – I would suggest this explanatory sentence be moved closer to the top of the paragraph. 4. Page 5 line 148 – where is the Iberian Peninsula? 5. Page 6 line 208 – The Figure 5 caption does not reflect what this sentence here days 6. Page 6 line 212 – I would like the 4 factors first, describe their relevance and then go into the results. 7. Page 7 line 240 – this should be said above to explain the figure 8. Page 8 line 276 – remove "On the other hand" its not really necessary. 9. Page 9 line 297 – "exploited" I think you mean explored? 10. Figure 1 – its hard to read. I suggest you average the data up to daily and then plot that so you don't have the significant noise 11. Figure 2 – In the figure caption, please include details about each planel. 12. Figure 4 –Do you need to repeat normalized level in each plot? I think more explanation in the figure caption would be appreciated 13. Figure 6 – I am not a fan of acronyms in figure captions without it having been written out. 14. Figure S1 – really hard to read the left hand plots. Is is necessary to have all the information in each plot? If it is maybe put it in a table elsewhere? I think you mean units and not unity.

Please also note the supplement to this comment:
https://www.atmos-chem-phys-discuss.net/acp-2020-102/acp-2020-102-RC2-

supplement.pdf

---

## Author Comment (AC1) · 6 May 2020

Author responses to Referees comments on the paper "Source apportionment of atmospheric mercury in the remote marine atmosphere: Mace Head GAW station, Irish west coast" by Custodio et al.

The authors are grateful for the referees for the comments, which improved the manuscript considerably. We provide point-by-point responses below in **bold,** and new statement in **red**.

**Referee #1**
Mace Head, GAW station provides valuable long-term observational data for atmospheric mercury in a coastal region in middle-latitudes. The data has been extensively interpolated for source appointment and atmospheric trend attribution. In this study, the authors utilized a receptor modeling technique for source apportionment that involves other chemical atmospheric trace species and meteorological data. This is a new addition to mercury trend analysis and the conclusions are generally reasonable. Clearly it merits publishing in ACP, but not in the current form. My major concern is the organization of the article. The authors made conclusions and speculations all through the results and discussion section. But some of them are not fully supported and seem hasty. I suggest reorganizing the paper to separate the result and discussion sections. By this mean the author can first present all the results, and then interpret them, especially their interactions as they are so closely associated (e.g. the results of hourly, monthly, and annual cycles and their associations with other chemical tracers and meteorological data). My detailed comments are as follow:

**R: The authors understand that such organization structure (separating results from discussion) could help the reader finding the manuscript results. The reason for do not split the section in results and discussion subsection, comes from the perspective of the told history. The authors prioritize the connection and insertion of results in the context of atmospheric mercury understanding. Moreover, splitting the aforecited section in effect of physical and qualitative analysis with measured parameters and a quantitative apportion with vector strength analysis can helps more in the structure and facilitate the understanding the results and research meaning.**

Line 136-137: this conundrum has an explanation in Zhang et al. PNAS, https://doi.org/10.1073/pnas.1516312113.

**R: A new statement, one paragraph was added with the following edits:**

**Line 144 to 149: "This conundrum related to increasing global emissions on one hand and measured declines in atmospheric mercury is discussed by Zhang et al. (2016). They state that the inventories do not account for the decline in the atmospheric release of Hg from commercial products, and do not properly account for the change in $Hg^0$/$Hg^{II}$ speciation of emissions from coal-fired utilities after implementation of gases emission controls"**

Line 167-169: It's not clear how the standard electrode potential or the kinetic coefficient of reactivity is translated to the conclusion that "Hg0 is quite a stable vapor gas, and a significant daily mass depletion by photooxidation is very unlikely".

**R: The stability of a chemical element is measured by his thermodynamic oxidation potential, which will reflect, or even determine its ΔH. Since thermodynamic is not the only control factor of reaction,**

including oxidation, they can be catalyzed, as happen with mercury in the atmosphere. The kinetic coefficient mentioned above consider ·OH, Br·, $O_3$, organic radical catalysis.

In order to improve, the authors reworded the sentence to "$Hg^0$ is a chemically relatively inert towards gas-phase oxidation, and a significant daily mass depletion by photooxidation is very unlikely. (Lines 182-183)".

Figure 4: wind direction has no y-axis.

R: It was normalized, we understand that radians normalization is not intuitive. A new axis with a new metric normalization was prepared for the plot of wind direction.  A new plot with a new metric for wind direction was prepared, figure 4 in the revised MS

Line 194: A specie with a lifetime of _0.5 yr is not a short-lived one.

R: We appreciated the comment. The main point here is the $CHCL_3$ has a shorter lifetime than mercury. The sentence was reworded to "…because $CHCl_3$ is a shorter-lived species (lifetime ~0.5yr"; line 206.

Section 3.1. The authors made some conclusions in this section, e.g. line 169-170 and 199-201. These conclusions seem unreliable and hasty. Why not waiting after presenting the PMF results?

R: Line 169-170 we simply refer to the well-known stability of elemental mercury against oxidation based on physical-chemical properties of the element (kinetically and thermodynamically), no new fundamental insights on atmospheric chemical reaction is presented.
About Line 199-201, the transport discussion is not propelled by PMF solution, but from the chemical species loaded in. The sentence is supported by the closer pattern between mercury and carbon monoxide as a well-known primary pollutant. The wind pattern presented in this section also supports such statement. The increase of TGM in winds from continental areas, winds with a high load of primary pollutants endorse and corroborate the PMF solution but is not a conclusion from the solution.
The sentence was rewrote as "Figure 3, 4 and S3, show that the seasonality in TGM observed in Mace Head is closely related to other species linked to primary sources and can be explained by transport from continental areas", now line 211-212.

Figure 5: The histogram of reconstructed value is not helpful. I suggest showing that of error.

R: The figure presents correlation, histograms for observed and reconstructed and 25/75th and 10/90th quantile values, plotting the error. The figure legend was improved in the revised MS in order to clarify the histogram meaning and help the reader identifying the error.
A new statement "Correlation among total elemental mercury measured and mercury reconstructed by the PMF solution and conditional quantiles plot showing the difference between PMF solution and observation. The observations are split up into bins according to correspondent reconstructed value. The median prediction line together with the 25/75th and 10/90th quantile values are plotted together with a line showing a "perfect model". It also shown is a histogram of reconstructed values (shaded grey) and a histogram of observed values (shown as a blue line)."

Line 224-233: "atmospheric acidification"? Is it actually "atmospheric oxidation"? Also, this paragraph reads very confusingly with so many turns around.

**R: $CO_2$ is an acid oxide, the ΔG of metals (Gibbs free energy) are strongly affected by acidification, reducing $E_H$, for example the Pourbaix diagram. Our point here is, it is possible that the increasing of $CO_2$ concentration in the atmospheric potentially affects the $Hg^0$ lifetime.**
**The paragraph was reformulated based the comments 16 from Referee #2. The aforementioned sentences now appear reworded as "**Another possible explanation for the declining trend may be the $Hg^0$ atmospheric life-cycling reduction due to atmospheric acidification caused by $CO_2$ increase and its potential (E°) to force elemental mercury oxidation**" line 238-240.**

Line 215-233: I suggest cutting the length of such speculations, they are very long and basically a review of past results. What new information is revealed by the author's own data and analysis?

**R: We appreciate and understand the reviewer's concern, it is important to be consistent, pragmatic, and mainly be supported by evidence. However, mercury sources and fades are complex and not well described yet. In the aforementioned paragraph we cite other statements and scientific premises to support our argument and finding.**
**The paragraph was reformulated also based in the comment above and comment 16 from Referee #2. The new statement is in line 229-250 in the revised MS: "**The PMF results show a statistically significant decrease in the baseline factor that could explain almost all of the trend changes in atmospheric mercury. This suggests a major decrease of anthropogenic inputs on a global scale. Slemr et al. (2011) reported a worldwide trend of atmospheric mercury, showing an equally strong decrease in the northern and southern hemispheres, which supports the argument of baseline-driven TGM decline.

According to Streets et al. (2011), anthropogenic Hg emissions in the USA and Europe decreased by 20% and 40%, respectively, from 1990 to 2008. However, emissions on a global scale, particularly from East Asia, are poorly reported (UN, 2018), even for most of the countries that are signatories of Minamata convention (UN, 2019). Moreover, the total emissions from small scale artisanal gold mining are highly uncertain estimates.

Another possible explanation for the declining trend may be the $Hg^0$ atmospheric life-cycling reduction due to atmospheric acidification caused by $CO_2$ increase and its potential (E°) to force elemental mercury oxidation. As reported by Slemr et al. (2011) and references therein, an increase in the atmospheric reactivity can induce large decreasing trends in the concentration of many long-lived substances. Clerbaux and Cunnold, (2007) did not observe lifetime changes for halogenated and other greenhouse gases, however, changes in oxidation rates of elemental mercury in the atmosphere could follow different kinetics. Furthermore, the increasing UV radiation and the shifting solar radiation to shorter wavelengths could also intensify the oxidation of elemental mercury into $Hg^{2+}$ (IPCC, 2007; Qureshi et al., 2010). Based on a global box-model of mercury biogeochemical cycling Streets et al. (2011) present a trend of atmosphere mercury from 1850 to 2008 showing the increase of $Hg^{2+}$ in the atmosphere in recent decades**."**

Line 260: It's risky to call this component as oceanic contribution as i) the fraction is very low; and 2) the baseline fraction may contain a contribution from the global ocean evasion fluxes. I

would suggest using the term "nearby ocean contribution". Many orphan sentences throughout the article. I suggest combine them with near by paragraphs.

**R: Thanks for the comment, this is true. As CHCl₃ fades, the factor signed by this species loses load of mercury; the mercury lost in the ocean factor is reloaded in the baseline factor.**

**We are not convinced to call the factor "nearby ocean" since the lifetime of chloroform is not so short allowing the specie to be transported for long distances. For example, air masses from North America can reach Mace Head within 96 hours and even less.**

**We are rewording the sentence to "This study shows an oceanic contribution (based on ocean factor solved by PMF) of 13% (0.17 ± 0.07 ng m$^{-3}$) to atmospheric TGM at Mace Head station." Line 279-280 in the revised MS.**

**Anonymous Referee #2**

Overall this is a good manuscript. The results and analysis are valuable additions to the scientific community. I have provided some technical suggestions to hopefully improve the manuscript and some editorial suggestions to improve the readers experience. My recommendation is that this manuscript be accepted with revisions. Technical suggestions: 1. Page 2 line 48. You use data from January 2013 to March 2018. I would suggest that when reporting annual trend data that you do not include 2018 in that data given that you do not have a full year and it may skew the results. Of course monthly reporting works but be mindful with annual reporting. 2. Page 3

**R: The matrix-vector introduced in PMF had time resolution of hours, 2018 contributed, roughly, with 2160 equations for the positive factorization. The manuscript has been revised in order to make clear that the time series is only until March 2018 and does not include all 2018 months.**

**New statement: "Account the more recent years (1996 to 2018, March), this decline continued with approximately 0.025 ± 0.04 ng m$^{-3}$ yr$^{-1}$, figure 2." Line 134-136 in the revised MS.**

line 72 – please add in a description of how the data was quality assured. What is the level of completeness of the data used in the analysis. This is very important when the data is compared to other data sets.

**R: Thanks for the comment. Some level, instrument failure is inevitable, they are susceptible to malfunctions that can result in lost or poor-quality data. Some data quality control steps are taken to minimize the risk of loss and to improve the overall quality of data. The validation process, In order to ensure data reliability and comparability of Mace Head mercury data follows the GMOS-Data Quality Management (G-DQM) protocol described by D`Amore et al. (2015). This statement will be reworded in the aforementioned paragraph.**

**New statement in the revised MS, line 74-79: "At some some level, instrument failure is inevitable, they are susceptible to malfunctions that can result in lost or poor-quality data. Some data quality control steps are taken to minimize the risk of loss and to improve the overall quality of data. Validation process, In order to ensure data reliability and comparability of Mace Head mercury data follows GMOS-Data Quality Management (G-DQM) protocol described by D`Amore et al. (2015) through a human check at Helmholtz-Zentrum Geesthacht."**

D'Amore, F., Bencardino, M., Sergio Cinnirella, S., Sprovieria, F., Pirrone, N.: Data quality through a web-based QA/QC system: implementation for atmospheric mercury data from the Global Mercury Observation System. Environmental Science Processes Impacts, 17, 1482–1491. DOI: 10.1039/c5em00205b, 2015.

The G-DQM system is a web-based tool aimed to control data quality that has been specifically developed to ensure data comparability among atmospheric mercury datasets collected within the GMOS network. Its application to three years of data allowed a very detailed analysis for each Tekran analyser used in the network. This centralized tool gave a fast and general overview of the analyser behaviour, and a rapid check of data quality. The fags adopted to tag values within datasets allowed us to understand issues occurring frequently and noticeably affecting data quality. The analysis performed here by means of the G-DQM on the GMOS network should be considered preliminary, since the site operator approval step is necessary to Analize the validation process through a human check. However, the results presented here provide an important first assessment of the mercury data acquired with the on-going GMOS stations and give important feedback for future instrument management and maintenance guidelines that could be taken into account in further development of mercury-oriented monitoring networks. G-DQM has been specifically designed to give rapid feedback on monitoring of atmospheric mercury based on the Tekran instrument, and is now being expanded to include the mercury analyser manufactured by Lumex, following ad-hoc SOPs. Further progress will also include an inter-comparison with existing systems aimed to quality assure and control mercury datasets. Apart from mercury, the amount of environmental data in general is expected to increase rapidly in the coming years, thus there is an increasing need for automated, platform-based methods to check and correct data to ensure that datasets provided to various end users are of highest quality

**R: The authors totally agree and are engaged and committed with G-DQM. The mercury laboratory from Helmholtz-Zentrum Geesthacht has extensive experience with atmospheric mercury measurements, working with Tekran QA/QC since 1993 and integrate the operator approval step of GMOS-Data Quality Management.**

3. Page 3 line 75 - please mention whether there was a filter on the outside inlet and if there is a rain shield etc. This is important again for comparison with other data sets. You can refer to other papers as appropriate if described there. 4. Page 3 lines 89-97 – I don't understand this very much. Can you give a small sentence with what this analysis will provide to the data in layman's terms?

**R: The authors are taking the comment and reword the sentence. Lines 80-84 in the revised MS:** "The air-sampling inlet is located on a tower at 10m agl (18m amsl) with a rain shield only. Air is sampled at a flow rate of 1 L/min through unheated PTFE tubing (1/4" O.D.) to the instrument, which is located in an air-conditioned laboratory. As reported by Weigelt et al. (2015), a PTFE pre-filter (pore size 0.2 mm) at the inlet of the instrument protects the sampling cartridges from contamination by particles."

5. Page 3 line 101 – it would be interesting to see the information of the accuracy of each of the analytical species to see where the largest error occurs. If doable, a table in supplemental information?

**R: The sentence will be reword in order to give reference to analytical accuracy and method description reported by Stanley et al. (2018).**

**Line 109-111 in the revised MS:** "The uncertainty input in the matrix was estimated based on the analytical accuracy of each individual species reported in Stanley et al. (2018) and Weigelt et al (2013)."

Stanley, K. M., Grant, A., O'Doherty, S., Young, D., Manning, A. J., Stavert, A. R., Spain, T. G., Salameh, P. K., Harth, C. M., Simmonds, P. G., Sturges, W. T., Oram, D. E., and Derwent, R. G.: Greenhouse gas measurements from a UK network of tall towers: technical description and first results, Atmos. Meas. Tech., 11, 1437–1458, https://doi.org/10.5194/amt-11-1437-2018, 2018.

Weigelt, A., Temme, C., Bieber, E., Schwerin, A., Schuetze, M., Ebinghaus, R., and Kock, H.H.: Measurements of atmospheric mercury species at a German rural background site from 2009 to 2011 – methods and results. *Environ. Chem.*10, 102–110, 2013

6. Page 3 line 104 – "the method provides a better solutions" – what do you mean by that? Solution to what? Be specific.

**R: Matrix factorization is an algebraic calculation assessing the strength of the vector with physical meaning (sources). Afterwards, what we obtain is a mathematical solution describing the observation, and reporting the differences in a degree of confidence. The "better solution" basically comes from a residual analysis in a context of the physical meaning of factors, beyond the creation of factors rather the restrict it based on constrained inputted source profile.**

**The sentence was reworded to "**The method provides a better qualitative solutions and time resolution of sources than principal component analysis (PCA) (Huang et al., 1999) or chemical mass balance (CMB) since PMF can generate source profiles ("learning algorithm") and let input of uncertainties which allow individual treatment of matrix elements.**" Line 112-115 in the revised MS.**

7. Page 4 lines 106-120 – The reader would benefit from a short explanation in simple terms on what one gets from this type of analysis.

**R: It is true that this type of analysis is well established and described in the literature, however, there are considerations in the calculation that have to be reported in order to make the analysis reproducible by others. Moreover, as far as the authors know, there is no other publication making use of a positive rotated factorization matrix to solver atmospheric mercury probabilistic mass function.**
**The authors are stating the benefit from this type of analysis in line 112-115 in the revised MS "**The method provides a better qualitative solutions and time resolution of sources than principal component analysis (PCA) (Huang et al., 1999) or chemical mass balance (CMB) since PMF can generate source profiles ("learning algorithm") and let input of uncertainties which allow individual treatment of matrix elements."

Page 4 line 125 – what do you mean "central tendancy"?

**R: Actually is "central tendency" and mean the central tendency of a statistical probabilistic distribution often called average. * normalized distribution.**
**Line 132-133 in the revised MS "**Concentrations range from 0.9 to 3.3 ng m$^{-3}$, displaying a central tendency of 1.3 ± 0.2 ng m$^{-3}$."

9. Page 4 line 128 – does this trend include the Jan – march 2018 data and should that be included?

**R:  The sentence is being reword to "**Account the more recent years (1996 to 2018, March), this decline continued with approximately 0.025 ± 0.04 ng m$^{-3}$ yr$^{-1}$**,"**

10. Page 4 line 127-128 – any thoughts on why the decline is so consistent? Have you looked to see if there is a change over time (the data from 1995 is available) to see if the rate of decline changes year over year or periodically? Maybe look at the trends in 5 year trends similar to how emissions are reported (i.e. each 5 years)?

**R: A short period trend can be misleading since it can be affected by inter annual differences, seasonality (as start the series in autumn/winter and finishing in spring/summer) as well starting, or finishing in an El Niño year can potentially constrain the trend. About the consistency between trend reported by Ebinghaus and the present study, it is one of the drivelines in the manuscript`s discussion, mainly concerning sources and fate of mercury in the atmosphere. Moreover, changes in the trend over the 21 years period is plotted in figure 2.**

11. Page 5 line 164 – in the reported studies of diurnal cycles, were they coastal sites like Mace Head?

**R: The point in the aforementioned sentence is the absence of a well-established statement for photochemical oxidation of mercury driving its diurnal cycle. For example Kalinchuk et al. (2019) reported decreasing in mercury in the day time and Wu and Nair (2010) report an increase in mercury concentration in the day time. Our manuscript explore the effect of transport driving the diurnal variation as presented in figure 4 and discussed in section 3.1.**

12. Page 5 lines 167-169
–Photoxidation may be unlikely but you have to consider the atmospheric conditions that lead to enhanced photooxidation at various locations and not only consider the properties of $Hg^0$

**R: Thanks for the comment, this could lead to a long discussion that is not clear that can lead to somewhere. Atmospheric reactions are mainly of kinetic control, the principal catalytic agents are $\cdot OH$, $\cdot Br$, $O_3$, organic radicals, altogether will give a kinetic coefficient of reactivity $< 2.1 \times 10^{-12}$ $cm^3$ $molec^{-1}$ $s^{-1}$. We can play around with temperature, pressure, and concentration of those radials. Which atmospheric, or enhanced photooxidation could give a reactivity to diminish 2% of mercury in 12h! On the other side, assuming that $Hg^0$ could be significantly reduced by photooxidation in a diurnal cycle, that would imply in a major effect in the lifetime of this specie.**
**The sentence was reworded giving references of manuscript showing no significant seasonality of $Hg^0$ and $Hg^{+2}$.**
**Line 180-183 in the revised MS: "**With a underline{standard electrode potential} ($E^0$) of +0.85 V and a kinetic coefficient of reactivity of $<9.8 \times 10^{-13}$ to $2.1 \times 10^{-12}$ $cm^3$ $molec^{-1}$ $s^{-1}$, at 1 atm and 298 K (Khalizov et al., 2003; Shepler et al., 2007; Subir at al., 2011; Sun et al., 2016), $Hg^0$ is a chemically relatively inert towards gas-phase oxidation, and a significant daily mass depletion by photooxidation is very unlikely."

Line 264-272 in the revised MS: The wind patterns for the baseline, combustion and sea factors (discussed below) as displayed in the polar plot of Figure 6 indicate an interpretation of the PMF profile with "combustion" being mostly associated with easterly transport, "sea" being linked to north-westerly and south-westerly winds. The "baseline" factor does not correlate with any significant wind patterns.
Another hand, no seasonality was observed for the baseline factor, linking lower concentrations of mercury in the warm season mainly to transport or evasion patterns and less to deposition by

oxidation. For instance, no evidences of photooxidation increase in growing season was reported by Weigelt at al. (2013) which shows no significant seasonality in gaseous elemental mercury and gaseous oxidised mercury in a remote rural environment in Germany.

13. Page 7 line 214 - "the PMF results show a statistically significant decrease in the baseline factor" – I am not sure I see that in what is presented. Can you make it clearer please?

**R: The sentence was reworded.**
**Line 229-230 in the revised MS "**The PMF results show an expressive decrease in the baseline factor that could explain almost all of the trend changes in atmospheric mercury as observed in figure 1."

14. Page 7 line 220 – you mention the decreases but why not include the increases during that time as well.

**R: The contradiction is presented on pages 1 and 2 lines 37 to 46. The statement was improved in the revised MS based on a comment from reviewer 1 on line 138-143. "The increasing improvement of manufacturing processes involving mercury and regulations limiting the emissions from coal-fired power plants since the 1980s (Hylander and Meili, 2003; Pirrone et al., 2009) could be a possible reason for this observed decline at Mace Head. Jiskra et al. (2018) report the Hg$^0$ uptake by vegetation as an alternative mechanism for driving mercury depletion in the Northern Hemisphere atmosphere over the past 20 years."**
**Also in line 234-235:** "According to Streets et al. (2011), anthropogenic Hg emissions in the USA and Europe decreased by 20% and 40%, respectively, from 1990 to 2008." And lines 37 to 46 "The 2018 Global Mercury Assessment (UN, 2018) reveals that primary anthropogenic mercury emissions into the air are 2220 t/y, also indicating an increase of 20% from such sources in recent years. The 2018 UNEP Report (AMAP/UNEP, 2018) presents an inventory for the year 2015, which indicates that the greatest atmospheric mercury emissions resulted from combustion of fossils fuels, mainly coal. While mercury in the atmosphere is chemically inert, once released into this environment, all sources are of concern.
To compile a global assessment based on inventories requires a number of assumptions and generalizations (AMAP/UNEP, 2018). Several discrepancies are observed in the mass balance-based estimation: there can be large differences between estimates, and it is important to recognize that there are sources of error in all methods for estimating mercury emissions."

15. Page 7 line 221 – I don't think the point about countries being valid signatories is relevant. Initial reporting were only due in December 2019 and full reporting is only due 2021. So, this statement isn't fair under the context of the Minamata Convention.

**R: Actually, the Initial reporting were only due in December 2019 and full reporting is only due 2021 however the Minamata Convention on Mercury was approved by delegates representing close to 140 countries on 19 January 2013 in Geneva and adopted and signed later that year on 10 October 2013.**

16. Page 7 lines 224-232 – Maybe have a look at some trend data from speciated Hg monitoring and use that to comment on this paragraph.

**R: We really appreciate this comment. A new statement considering the increase of Hg$^{+2}$ in recent decades shows by Streets et al. (2011) will be considered.**

**Line 239-250 in the revised MS:** "Another possible explanation for the declining trend may be the Hg$^0$ atmospheric life-cycling reduction due to atmospheric acidification caused by $CO_2$ increase and its potential (E°) to force elemental mercury oxidation. As reported by Slemr et al. (2011) and references therein, an increase in the atmospheric reactivity can induce large decreasing trends in the concentration of many long-lived substances. Clerbaux and Cunnold, (2007) did not observe lifetime changes for halogenated and other greenhouse gases, however, changes in oxidation rates of elemental mercury in the atmosphere could follow different kinetics. Furthermore, the increasing UV radiation and the shifting solar radiation to shorter wavelengths could also intensify the oxidation of elemental mercury into Hg$^{2+}$ (IPCC, 2007; Qureshi et al., 2010). Based on a global box-model of mercury biogeochemical cycling Streets et al. (2011) present a trend of atmosphere mercury from 1850 to 2008 showing the increase of Hg$^{2+}$ in the atmosphere in recent decades."

17. Page 7 line 238 – But you have far more data available from this location, why not use that?

**R: We will use it. Source apportion of mercury is an ongoing project. We are working on data harmonization and increasing computer power.**

18. Page 8 line 244 – CHCl3 and CO are not mentioned in Figure 7

**R: Figure 7 shows the factors seasonality, the sea, and combustion factors are fingerprinted by CHCl$_3$ and CO respectively.**

**Line 257 in the revised MS:** "Moreover, seasonality observed in the factors fingerprinted by CHCl$_3$ and CO (Figure 7)".

19. Page 8 lines 244-255 – the explanations are a little all over the place. Perhaps a more organized discussion could be done here so the reader can follow more easily.

**R: The sentence was relocated in the previous paragraph. Line 237-260 in the revised MS:** "Moreover, seasonality observed in the factors fingerprinted by CHCl$_3$ and CO (Figure 7) should, however, be considered with caution because those short-lived species (CHCl$_3$ 4-5 months and CO 1-3 months) have lifetimes that vary by season, which can dampen mercury load into its factor during summer."

20. Page 8 line 264 – ": : :is forcing the atmospheric trends". Do you mean rather than emissions? If so, please state that so its clear

**R: The sentence was reworded, line 288-289 in the revised MS:** "They also argued, based on cruise data, that the decrease of oceanic emissions is forcing the atmospheric trend down."

21. Page 8 line 268 – maybe include the % here to stay consistent with the other 3 factors explained.

**R: In the significance reported for others factors this one will be ~0% as presented in figure 8. The sentence was reword.**

**Line 224 in the revised MS:** "One factor with a high load of $O_3$ and CO was found by the PMF solution which appeared to be irrelevant for the mercury mass balance, as its load was just 0.003 ng m$^{-3}$ (~0 %)."

Line 269, 0.57ngm-3 is very big! Can you put that in a percent and also offer more insight as to why.

**R: Really good comment. As reported in the manuscript, this accounts for 25% mercury mass for high concentration episodes. Among those episodes, we have for example biomass burning nearby the station. More insight into the high mercury concentration will be presented in follow-up manuscript presently under preparation.**

 **Editorial suggestions:**
 1. Page 2 line 56th. It's a little odd that you don't give this factor an identification as you have done for the others. I would suggest naming it something relevant to what its looking at rather than saying 4th factor which doesn't define it at all.

**R: We agree, however despite having a high load of ozone and carbon monoxide which could link it to anthropogenic source it was not possible to present a factor profile (chemical species load in the factor) that really seems a singular source coming from a specific spot or region. The point is, the number of days with high concentration is not so many, more equations can be necessary to strengthen this vector. Labeling it as a 4th factor could seem more conservative.**

2. Page 3 line 81 – remove "furthermore" this adverb is not appropriate here.

**R: It was reworded. Thanks,**

 3. Page 3 lines 102-104 – I would suggest this explanatory sentence be moved
closer to the top of the paragraph.

**R: Suggestion taken, sentence reworded. Thanks**

4. Page 5 line 148 – where is the Iberian Peninsula? (is that a joke ?)

**R: The Iberian Peninsula is the South-west European region that's most associated with the countries of Spain and Portugal.**

5. Page 6 line 208 – The Figure 5 caption does not reflect what this sentence here
days

**R: Figure 5 caption was reworded and improved based on the comment from the first reviewer.
Statement in the revised MS:** "Figure 5: Correlation among total elemental mercury measured and mercury reconstructed by the PMF solution and conditional quantiles plot showing the difference between PMF solution and observation. The observations are split up into bins according to correspondent reconstructed value. The median prediction line together with the 25/75th and 10/90th quantile values are plotted together with a line showing a "perfect model". It also shown is a histogram of reconstructed values (shaded grey) and a histogram of observed values (shown as a blue line)."

6. Page 6 line 212 – I would like the 4 factors first, describe their relevance and then go into the results.

**R: Section 3.2 was reworded based on this comment.**
**Line 216-226 in the revised MS:** "Figure 1 shows the set of four factors reconstructing atmospheric mercury concentrations obtained from the PMF solution. As reported by Henry (1991), the first set of natural physical constraints of the system to be considered in any approach for identifying and quantifying source mass contributions must be the reconstruction of the original data set by the algorithm—that is, the solution must explain the observations. Figure 5 shows that the sum of the predicted elemental mass contributions for all sources is almost the same as the total TGM measured. Lower reconstruction performance was observed in particular for concentrations higher than 2 ng m$^-$$^3$, which make up 0.44% of the observations. One factor with a high load of $O_3$ and CO was found by the PMF solution which appeared to be irrelevant for the mercury mass balance, as its load was just 0.003 ng m$^{-3}$ (~0 %). However, for atmospheric mercury concentrations higher than 2 ng m$^{-3}$ this factor had a load of 0.57 ng m$^{-3}$, and was labeled as fourth factor."

7. Page 7 line 240 – this should be said above to explain the figure 8.

**R: Section 3.2 was reworded and the sentence relocated.**
**In the revised MS the sentence is in line 264**

Page 8 line 276 – remove "On the other hand" its not really necessary.

R**: The sentence was reworded**
**Line 294 in the revised MS:** "A decrease in mercury is observed in the factor with high loading of long-lived species such as CFCs."

Page 9 line 297 – "exploited" I think you mean explored?

**R: We intended to say exploited, which mean that it was used to take advantage of.**

10. Figure 1 – its hard to read. I suggest you average the data up to daily and then plot that so you don't have the significant noise

**R: The average will dump variance and constrain significance. In order to build a solution based on data with a daily resolution, could be necessary a period longer than 5 years.**
**The graphic resolution was improved afford be in order to be easily afford scaled by the reader.**

11. Figure 2 – In the figure caption, please include details about each planel.

**R: The figure caption was improved.**
**The Statement in the revised MS:** "Figure 2. Time series decomposition of TGM (monthly averages) measured at Mace Head from 1996 to February 2018. From top to bottom it is presented the monthly time series followed by the patterns of deconstructed components, trend, seasonality and radon. * TGM in ng m$^{-3}$."

12. Figure 4 –Do you need to repeat normalized level in each plot? I think more explanation in the figure caption would be appreciated

**R: Figure 4 was improved based on comments of both reviewers, the normalized level was removed from plots on the left. It was add a secondary axis for wind dir.**

**The caption in the revised MS: "**Figure 4: Diurnal cycle and seasonal cycle of mercury and species loaded in the PMF matrix. The shaded areas are the 95% confidence intervals in the mean. *Wind direction is normalised with west 90° as -1 and east (270°) as 1.**"**

13. Figure 6 – I am not a fan of acronyms in figure captions without it having been written out.
14. Figure S1– really hard to read the left hand plots. Is is necessary to have all the information in each plot? If it is maybe put it in a table elsewhere? I think you mean units and not unity.

**R: We appreciate the comments, the figures and captions were reassessed.**

**We thank the two referee for the encouraging comment! With the joint effort from the authors, we edited the manuscript to improve the quality as a whole.**